# Classification of current density vector map using transformer hybrid residual network

**Lihui Zhu[1,2], Yunfeng Yang[1], Wenyue Yu[1], Qingxin Zeng[1], Zhan Zheng[1]\*, Qiang Lin[1]\*, Xiaohu Zhao[3]\*, Zhenghui Hu** [1]\*

**1** School of Physics, Zhejiang University of Technology, Hangzhou, Zhejiang, China, **2** Zhengshi Middle School of Ningbo, Ningbo, Zhejiang, China, **3** Department of Imaging, The Fifth People's Hospital of Shanghai, Shanghai, China

\* zhan@zjut.edu.cn (ZZ); qlin@zjut.edu.cn (QL); xhzhao999@263.net (XZ); zhenghui@zjut.edu.cn (ZH)

## Abstract

The classification of the current density vector map (CDVM) reconstructed from magnetocardiogram (MCG) is an important indicator for assessing cardiac function and state in clinical diagnosis. Given the limited widespread application of MCG, research on CDVM often encounters challenges such as scarcity of data and difficulties in judgment. There is growing interest in computer-aided methods to assist physicians in analyzing cardiac cases using CDVM. This paper proposes a deep learning-based approachto classify the CDVM. To address the issue of insufficient processed MCG data, data augmentation is carried out by adding noise, making predictions based on auto regressive integrated moving average (ARIMA) model, and utilizing interpolation methods. A transformer hybrid residual network is then employed to classify the CDVM across categories 0 to 4, with transfer learning incorporated into the network structure to initialize model parameters, and the self-attention mechanism of the Transformer enhancing the feature extraction capability. This method achieved a classification accuracy of 97.52%, outperforming previous deep learning approaches, exhibiting both high precision and efficiency. Furthermore, its high scalability ensures that it will continue to meet the evolving needs of physicians as CDVM datasets undergo continuous expansion.

## Introduction

For early detection and preliminary diagnosis of heart diseases, magnetocardiogram (MCG) is a valuable tool [1–3]. The MCG system is capable of precisely measuring the extremely weak magnetic field (in the range of $10^{-12}$ to $10^{-15}$ T) [4] produced by the current flowing within myocardial fibers during cardiac activity, thereby obtaining temporal and spatial distribution information regarding the electrical activity of the heart. MCG demonstrates numerous advantages [5,6], most notably as a non-invasive diagnostic tool [7,8]. Moreover, compared to Electrocardiogram (ECG) [9–11]. MCG offers enhanced objectivity and quantitative precision, allowing for the

**Data availability statement:** All relevant data are within the paper and its Supporting information files.

**Funding:** This work is supported in part by the National Natural Science Foundation of China under Grant U2341246, in part by the National Key Research and Development Program of China under Grant 2018YFA0701400, in part by Zhejiang Provincial Natural Science Foundation of China under Grant LD22F050003, in part by Great Discipline of Shanghai Minhang District under Grant 2024MWDXK03, and in part by Zhejiang Provincial Natural Science Foundation of China under Grant LD25A040002.

**Competing interests:** The authors have declared that no competing interests exist.

detection of subtle cardiac phenomena that might go unnoticed with conventional ECG measurements [12,13]. Despite its limited widespread adoption and utilization, MCG has garnered recognition for its remarkable clinical diagnostic prowess [14–16]. For instance, in patients with coronary heart disease (who have a high risk of sudden cardiac death), MCG can detect significant changes in their heart's magnetic field compared to healthy individuals [17].

MCG is technically intricate [18], necessitating sophisticated signal processing and imaging techniques to extract meaningful information from the collected magnetic field signals. The current density vector map (CDVM), reconstructed from processed MCG data, effectively mirrors the state of the cardiac magnetic field at a given moment [19]. The CDVM demonstrates remarkable proficiency in diagnosing certain specific diseases, including ischemic heart disease and other heart failures that are associated with alterations in current flow within heart muscles [20,21].

As the utilization of CDVM as a clinical diagnostic tool for heart disease is still in its infancy, the interpretation of CDVM requires experienced professionals. However, relying solely on manual assessment is both time-consuming and lacking in efficiency.Therefore, computer-based diagnostic technologies hold significant value for clinicians [22,23].

Research on CDVM classification remains in its developmental stage with substantial scope for advancement. Early approaches like k-Nearest Neighbor algorithm by Udovychenko et al. [20] achieved accuracy rates of 60-90%. More recently, Tao et al. [24] proposed MCG-Net, combining CNN and Transformer architecture, which reached 87.0% accuracy. However, the classification accuracy still has room for improvement, particularly given the unique challenges posed by CDVM analysis: limited datasets, complex spatial-temporal patterns in current density distributions, and the need to capture both local current flow details and global cardiac magnetic field interactions [25].

Recent studies have shown promise in ResNet [26] or Transformer [27] architectures for cardiac analysis, with Wahid et al. [28] proposed a hybrid ResNet-ViT approach for ECG-based myocardial infarction detection and Peng et al. [29] achieving success in ECG-based arrhythmia detection by incorporating Transformer architecture. Although these methods have demonstrated success in ECG analysis, their direct application to CDVM classification poses challenges. CDVM's distinctive characteristics - its representation of cardiac current density distributions and the need to simultaneously analyze both microscopic current flow patterns and macroscopic field interactions - require a specialized architectural design that differs from conventional cardiac image or signal analysis approaches.

In this paper, a novel hybrid architecture is proposed specifically tailored for CDVM classification that synergistically combines ResNet and Transformer networks. The approach is innovative in three key aspects: (1) a customized feature extraction mechanism that captures CDVM-specific local current patterns through modified ResNet blocks, (2) an adapted Transformer architecture designed to model the unique global dependencies in current density distributions, and (3) Added transfer learning [30] strategy that addresses the limited availability of CDVM data while preserving the physical significance of current density features. Experimental results

demonstrate superior performance with a classification accuracy of 97.52%, significantly outperforming existing methods in CDVM-based cardiac diagnosis.

The remainder of this paper is organized as follows. The Methods section presents the proposed methodology, encompassing the preliminary data preprocessing approaches and introduces the proposed deep learning framework, along with detailed experimental procedures. The Results section evaluates and discusses the experimental results. The Conclusion section concludes the paper with a summary of findings and potential future directions.

## Methods

This study was approved by the Research Ethics Committee of College of Science, Zhejiang University of Technology (Reference Number: 2024 Scientific Research Expedited Review No.7). MCG data from 18 participants were analyzed retrospectively. The data were accessed on June 17, 2024, from routine clinical records with all personal identifiers removed prior to analysis. The research team had no access to any information that could identify individual participants either during or after data collection. Each case was assigned a random numerical identifier to ensure patient privacy while maintaining data integrity. The dataset only contained relevant MCG data and clinical outcomes. Informed written consent was obtained from all participants, as documented by the approved informed consent form included in the ethics review materials.

### Data collection and preprocessing

The MCG measurements were performed using a Pulsed Pump Rb Atomic Magnetometer system [31,32], which recorded signals from 49 channels arranged in a 7×7 grid with 4cm spacing. The system operates at room temperature with a sensitivity of 10 fT/$\sqrt{\text{Hz}}$ in the frequency range of 0.1-100 Hz. Three subjects were excluded due to excessive motion artifacts (n=1) and incomplete data collection (n=2), resulting in a final dataset of 15 volunteers (11 healthy subjects and 4 patients with heart disease).

After collecting multi-channel MCG signals, baseline drift may occur due to the operating principles of the magnetometer and external conditions. Therefore, it is necessary to perform baseline correction on the raw MCG data. After addressing the baseline drift in the magnetocardiographic signals, pure magnetocardiographic signals are obtained, which serve as the basis for constructing the MCG and CDVM. For the plotting of MCG, the cubic spline interpolation method is employed. For CDVM plotting, the electric field of the cardiac dipole model can be inverted using field gradient magnetic field measurement data, allowing the current source vectors of this magnetic field distribution to be calculated based on the magnetocardiographic measurement data. The specific process for original MCG signal processing and imaging is illustrated in Fig 1.

Human cardiac magneto signals and electrocardiographic signals exhibit similar morphological characteristics [33], primarily consisting of P wave, QRS complex, J point, ST segment, and T waves, as shown in Fig 2. The collected multi-channel MCG data are shown in Fig 3. Patients with heart disease exhibit ST-segment displacement compared to healthy individuals [34]. Therefore, the CDVM reconstruction focused on the ST-T segment, which is crucial for detecting cardiac abnormalities. The temporal sampling interval was set to 5ms, considering the clinical requirements for ST-T segment analysis temporal resolution, and computational efficiency.

Considering the variability of cardiac magnetic cycle signals across different samples, 40-50 CDVMs could be reconstructed for each sample during the reconstruction process. An example of reconstructing CDVM from MCG raw data is shown in Fig 4. In total, 866 CDVMs were reconstructed to serve as the raw dataset.

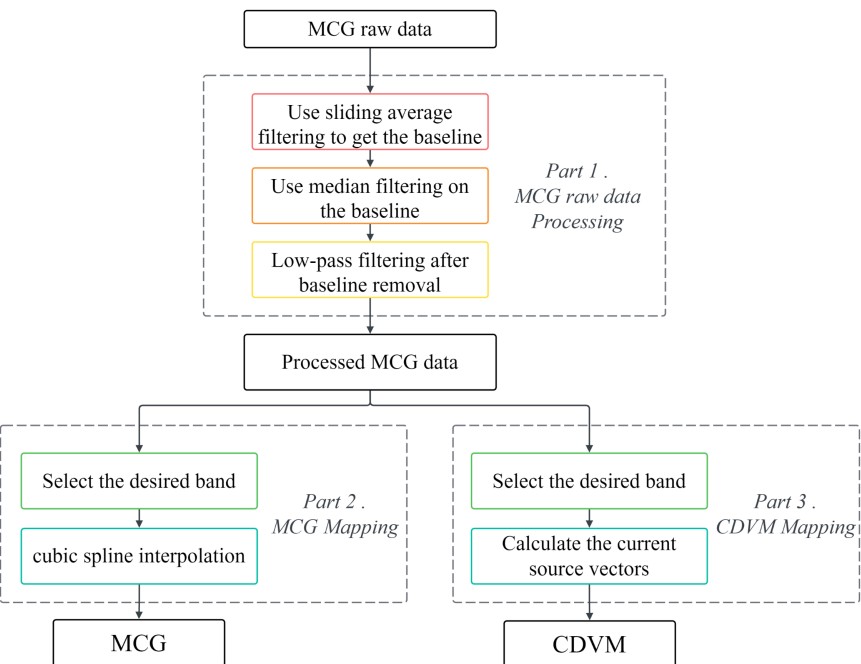

**Fig 1**. **CDVM imaging framework.** Processing of raw MCG signals and imaging of the current density vector map.

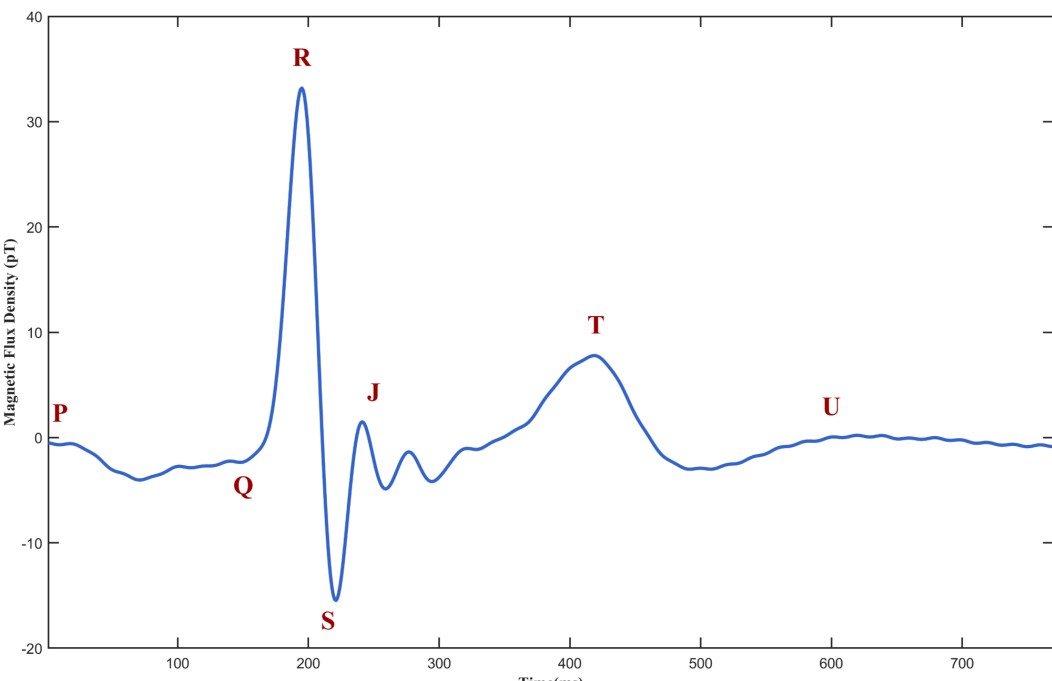

**Fig 2**. **Cardiac magneto signal.** A single-channel cardiac magneto signal waveform over one cardiac cycle, illustrating the typical morphology with P wave, QRS complex, J point, ST segment, and T wave.

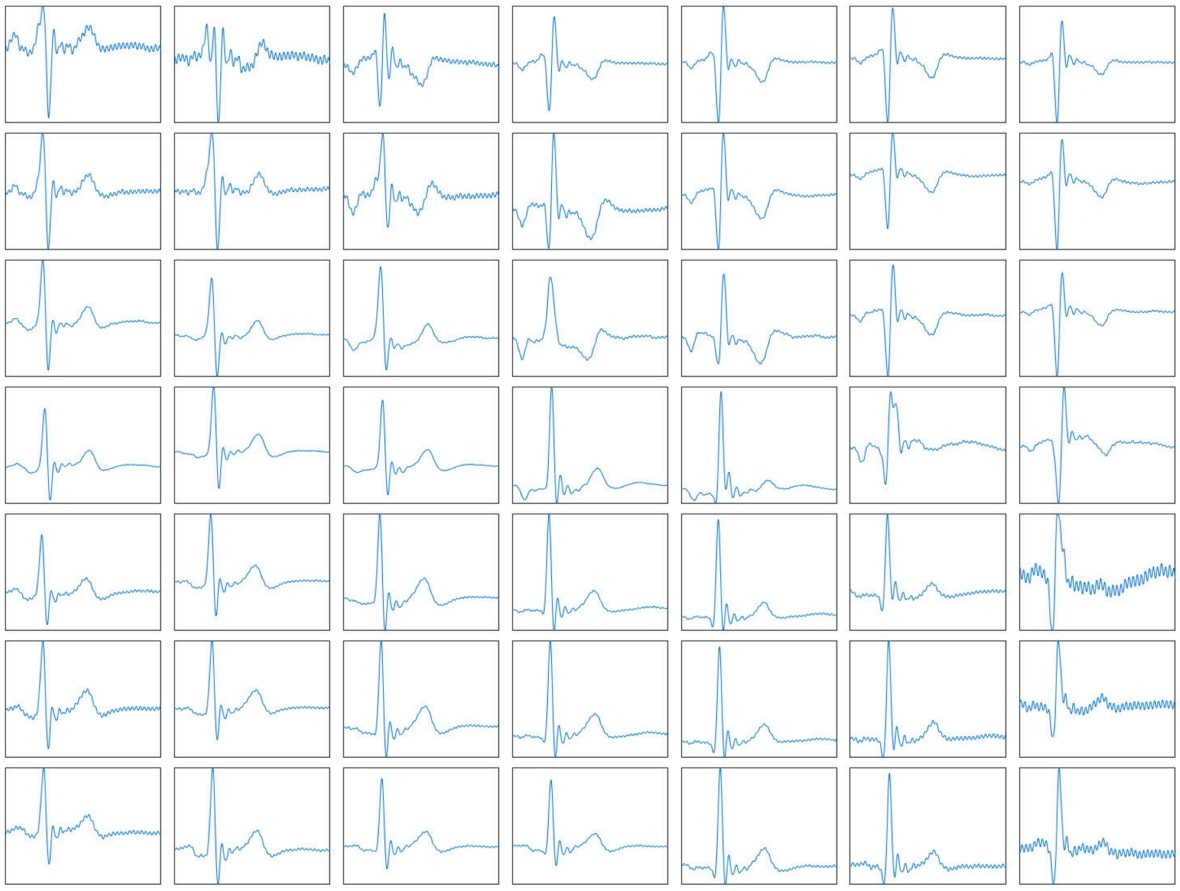

**Fig 3**. **Multichannel MCG data.** Multichannel MCG data from 49 channels.

## Dataset augmentation

Training a large neural network requires an extensive labeled dataset. However, due to the current lack of available large-scale public CDVM datasets, three complementary data augmentation strategies were implemented to expand the dataset while preserving the physiological characteristics of cardiac signals.

The first strategy involved systematic introduction of controlled noise to simulate real-world detection environments. Gaussian white noise was applied to simulate measurement uncertainty, random noise for environmental interference, and periodic noise to account for equipment-related disturbances. The second strategy utilized the auto regressive integrated moving average (ARIMA) model [35] for time series prediction, leveraging the temporal correlations in MCG data across channels. The model was optimized with parameters (p=1, d=0, q=1) determined through ACF and PACF analysis, configured to predict 5ms into the future with a segment length of 10ms. The third strategy employed multiple interpolation methods (cubic spline, polynomial, and linear) to generate intermediate MCG data points, taking advantage of the 49-channel MCG system's spatial-temporal characteristics and the continuous nature of cardiac signals.

Each augmented CDVM underwent rigorous assessment, including clinical expert validation, to ensure the preservation of diagnostic features. This multi-method approach effectively reduced overfitting while enhancing the model's generalization capability. An augmented CDVM example is shown in Fig 5.

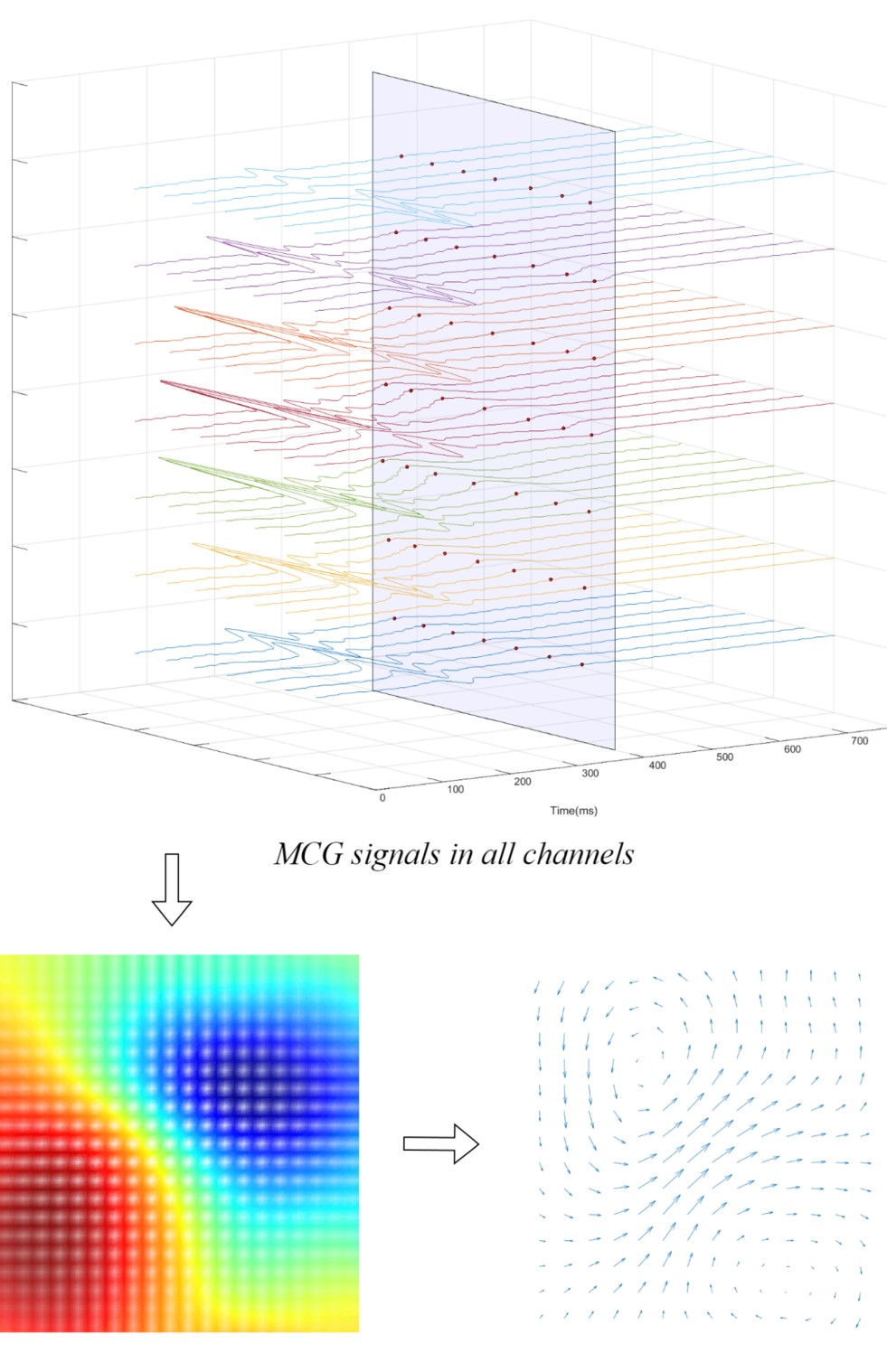

MCG signals in all channels

Magnetic Field Map          Current Density Vector Map

**Fig 4**. **Generation of current density vector map.** Raw 49-channel MCG signals were preprocessed (denoising and baseline correction) to plot the magnetic field map, and the current density vector map was reconstructed over the ST–T segment with a 5ms sampling interval.

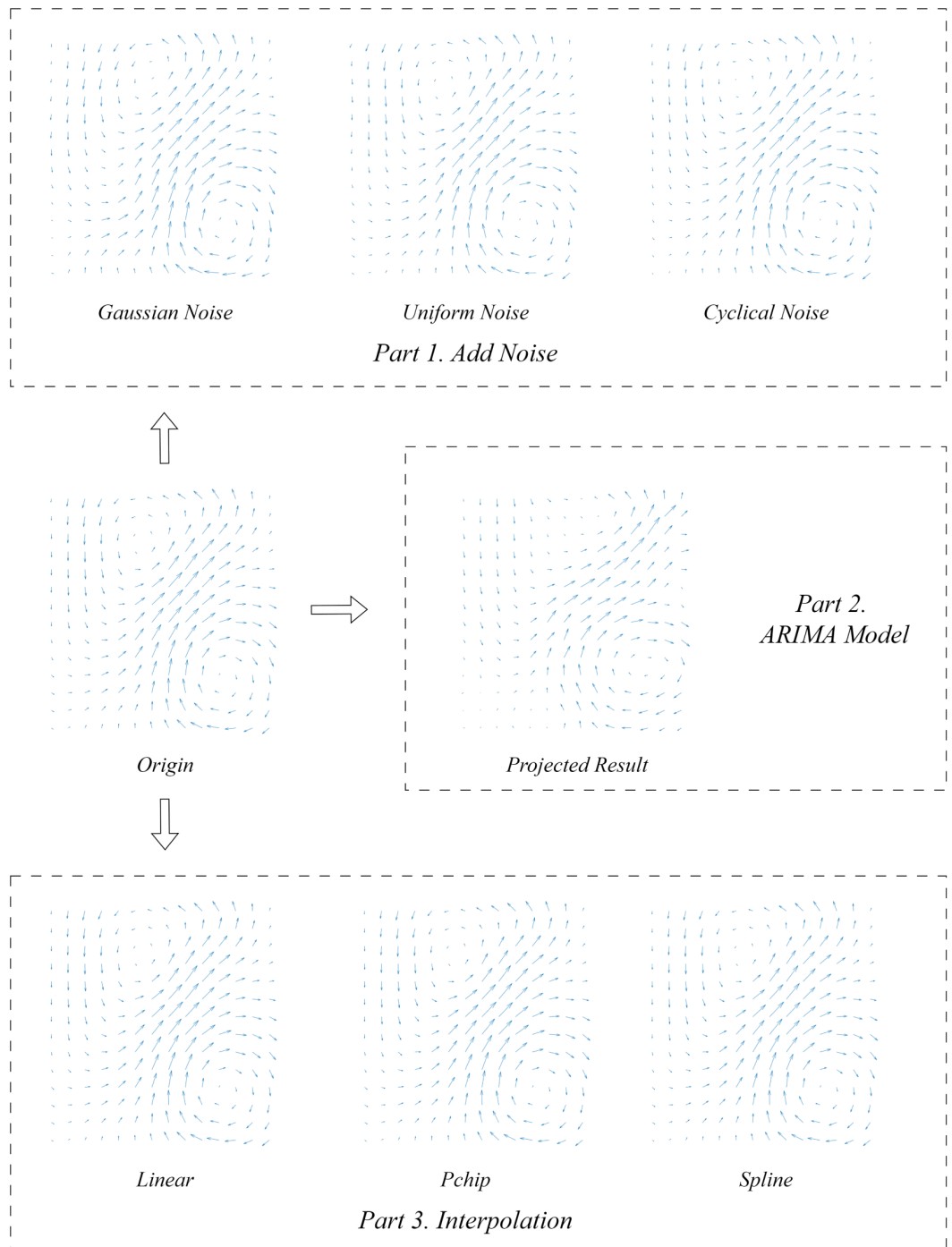

**Fig 5**. **Data enhancement.** Example of an augmented CDVM generated by noise addition, ARIMA prediction, and interpolation.

## Dataset construction and partition strategy

According to the standardized evaluation system established by clinical experts [21,36], CDVMs are classified into five grades (0-4) based on two critical parameters: the number of vortex structures and the direction of integrated current

vectors. Categories 0 and 1 represent normal patterns, category 2 indicates mild abnormality, while categories 3 and 4 signify severe abnormalities, as illustrated in Fig 6. Clinical diagnosis primarily relies on two quantitative indicators: the Average Classification of the Total Current Density Vector Maps (ACTM) and the Ratio of the Number of Abnormal Maps to the Total Number (RAM). A positive diagnosis is typically made when $ACTM \geq 3$ or $RAM \geq 0.3$, indicating a high frequency of abnormal CDVMs within specific time windows.

The dataset comprises recordings from 15 subjects (11 healthy individuals and 4 cardiac patients), yielding 4,330 CDVMs after augmentation. To ensure robust model evaluation while maintaining clinical relevance, a stratified 6:2:2 partition strategy was implemented for training, validation, and test sets. The training set consists of 9 subjects (7 healthy, 2 patients) with 2,598 CDVMs, providing sufficient representation of both normal and pathological cases for effective model learning. The validation and test sets each contain 3 subjects (2 healthy, 1 patient) with 866 CDVMs, enabling proper model optimization and unbiased performance evaluation respectively.

This partition strategy was carefully designed to maintain consistent representation of healthy and pathological cases across all sets while providing sufficient data volume for effective model training and validation. The approach preserves subject-level separation to assess true generalization capability and balances the need for robust training with reliable validation and testing. The distribution of CDVMs across categories (290 category 0, 380 category 1, 645 category 2, 780 category 3, and 2,235 category 4) was maintained proportionally in each partition to ensure representative sampling. Most importantly, this subject-wise separation, rather than random sampling or cross-validation, better reflects real-world clinical scenarios while providing a solid foundation for model evaluation with limited medical data availability.

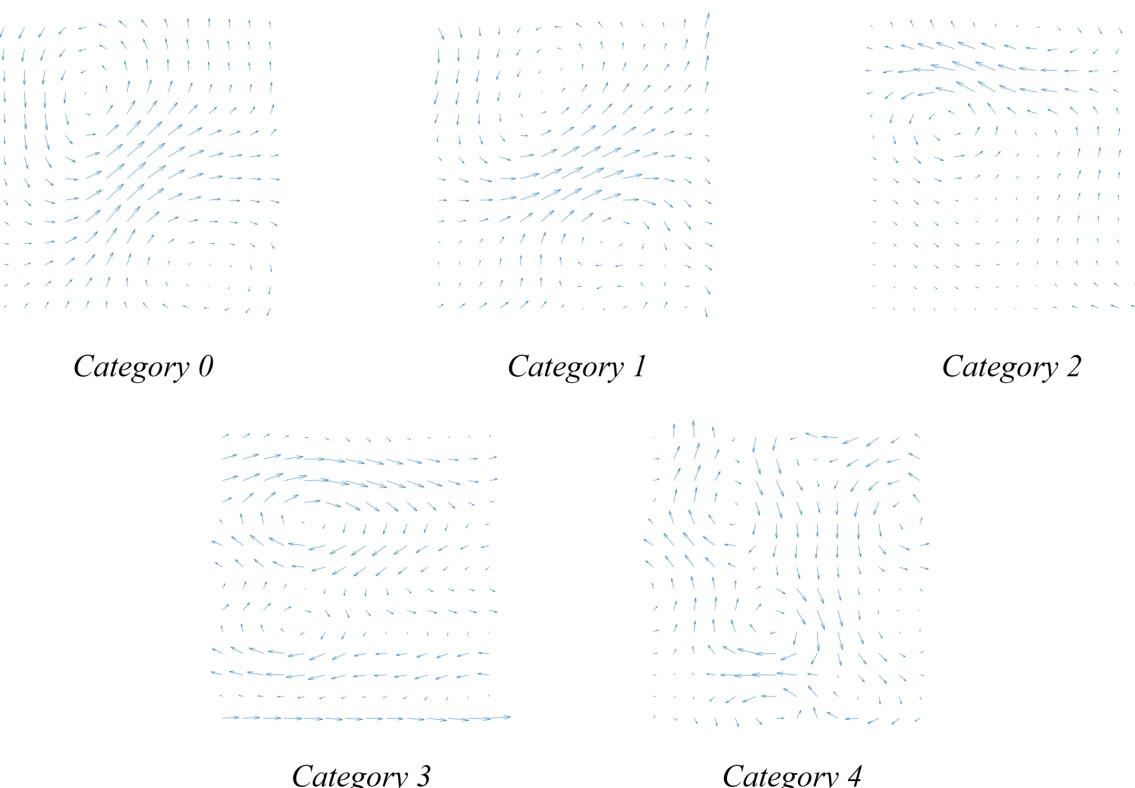

*Category 0*          *Category 1*          *Category 2*

*Category 3*          *Category 4*

**Fig 6. Classification system of CDVM.** Category 0–1 denote normal patterns, category 2 indicates mild abnormality, and category 3–4 signify severe abnormalities. Representative examples for each category are shown.

## Construction of neural network

The overall architecture of the proposed network is shown in Fig 7. The network consists of a ResNet-18 backbone for feature extraction, followed by Transformer encoder modules for capturing global dependencies, and finally a classification head for CDVM pattern recognition.

**Base network selection.** The selection of backbone network architecture is crucial for CDVM classification tasks. ResNet-18 was selected as the backbone network due to several key theoretical advantages: First, the residual connections effectively address the degradation problem in deep networks, enabling stable gradient flow during training. Second, the hierarchical feature representation aligns well with the multi-scale nature of magnetic field patterns, where both local current patterns and global field distributions need to be captured. Third, the moderate depth of ResNet-18 provides an optimal balance between model capacity and computational efficiency, particularly suitable for the given CDVM dataset scale.

Based on these considerations, ResNet-18 was selected as the backbone network. ResNet, introduced by He et al. [26], effectively addresses the degradation problem through its innovative skip connections, enabling stable gradient flow during training. The architecture provides an optimal balance between model complexity and feature extraction capability, with its 18-layer depth particularly suitable for the dataset size. Experimental results validate this architectural choice, as the ResNet model achieved significantly higher accuracy compared to both the MLP (50.64%) and basic CNN (57.72%) implementations, demonstrating superior feature extraction and classification capabilities.

**Transfer learning integration.** The selection of transfer learning [37] strategy is essential for the CDVM classification task, particularly considering the domain-specific dataset characteristics. Transfer learning with ImageNet [38] pre-trained weights is adopted for several key theoretical advantages: First, despite the domain gap between natural images and magnetic field visualizations, the low-level features (e.g., edges, textures, and basic geometric patterns) learned from ImageNet are fundamentally transferable to the task. Second, pre-trained weights provide a well-initialized starting point in the parameter space, which is crucial for avoiding poor local optima during training. Third, the knowledge transfer from a large-scale dataset helps mitigate the potential overfitting issues that could arise from the relatively limited domain-specific data.

**Transformer encoder integration.** Transformer encoder modules [27] were integrated into the architecture to address the long-range dependencies inherent in CDVM data. Although ResNet18 performs well in extracting local features, it has limitations in capturing the long-range relationships for analyzing current density patterns. The Transformer encoder architecture is adopted for several theoretical considerations: First, the self-attention mechanism enables dynamic modeling of global relationships, allowing the network to correlate features regardless of their spatial distance.

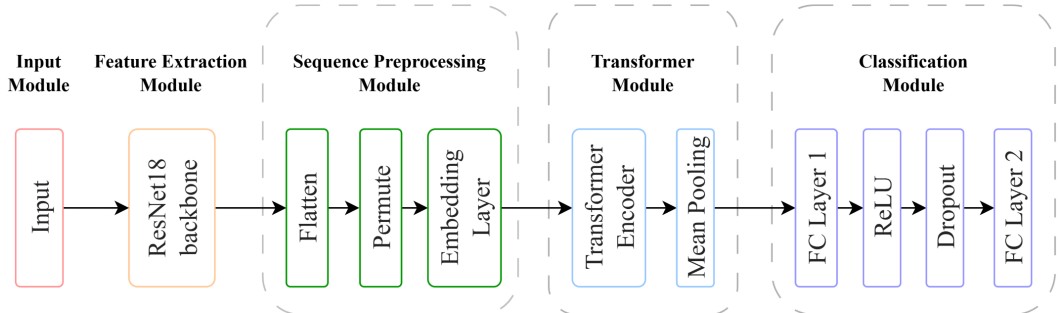

**Fig 7. Network structure schematic diagram.** The model consists of a ResNet-18 backbone for local feature extraction, a Transformer encoder for global context modeling, and classification heads for CDVM classification. Data preprocessing and augmentation are applied prior to training; arrows indicate the forward flow.

Second, the encoder's parallel processing of spatial information facilitates the capture of current distributions across the entire image space.

The implementation employs a standard Transformer encoder block consisting of multi-head self-attention (MSA) followed by a multi-layer perceptron (MLP), with layer normalization and residual connections. The self-attention mechanism computes attention weights between feature positions using scaled dot-product attention:

$$\text{Attention}(Q, K, V) = \text{softmax}\left(\frac{QK^T}{\sqrt{d_k}}\right)V \tag{1}$$

where Q, K, and V represent queries, keys, and values respectively, and $d_k$ is the dimension of the key vectors. The multi-head structure allows the model to jointly attend to information from different representation subspaces, enhancing the model's ability to capture various aspects of the current density patterns.

### Training strategy

For the CDVM classification task with limited samples, the selection of appropriate hyperparameters, optimizer, and prevention of overfitting are crucial aspects of model training. These optimization procedures were implemented to enhance the model's performance and accuracy.

**Hyperparameter optimization.** Bayesian optimization was employed for hyperparameter tuning, leveraging its efficiency in exploring high-dimensional parameter spaces. This approach utilizes probabilistic surrogate models to guide the search process, making it more efficient than traditional grid or random search methods. The optimization process focused on critical model parameters including learning rate, dropout rate, number of transformer layers, and weight decay coefficient. Through this systematic optimization strategy, the optimal configuration for the model architecture was efficiently identified while minimizing computational resources.

**Optimization.** Model optimization was performed using Adam optimizer, which combines adaptive learning rate adjustment with momentum-based gradient updates for efficient training convergence. Adam's ability to handle varying parameter scales and sparse gradients makes it particularly suitable for the hybrid architecture combining ResNet and Transformer components.

**Overfitting prevention.** To address the overfitting challenges in the model training process, several key regularization techniques were implemented. The primary strategy involves comprehensive data augmentation, including random cropping and horizontal flipping to enhance the diversity of training samples. The training process was stabilized through the incorporation of dropout mechanisms in the classification head and the application of weight decay regularization in the optimizer. Additionally, an early stopping mechanism was implemented to prevent overfitting. These complementary techniques work in concert to ensure model generalization despite the limited sample size of the CDVM dataset.

## Results

All experiments were conducted on the CDVM dataset using the PyTorch framework. Each model was trained for a maximum of 100 epochs with an early stopping patience of 5 epochs to prevent overfitting.

### Performance analysis of network architectures

To systematically evaluate the effectiveness of different network structures, three representative deep learning architectures were implemented and compared: Convolutional Neural Network (CNN), Multi-Layer Perceptron (MLP), and ResNet18. The experimental results, shown in Fig 8, reveal notable differences in classification performance among these

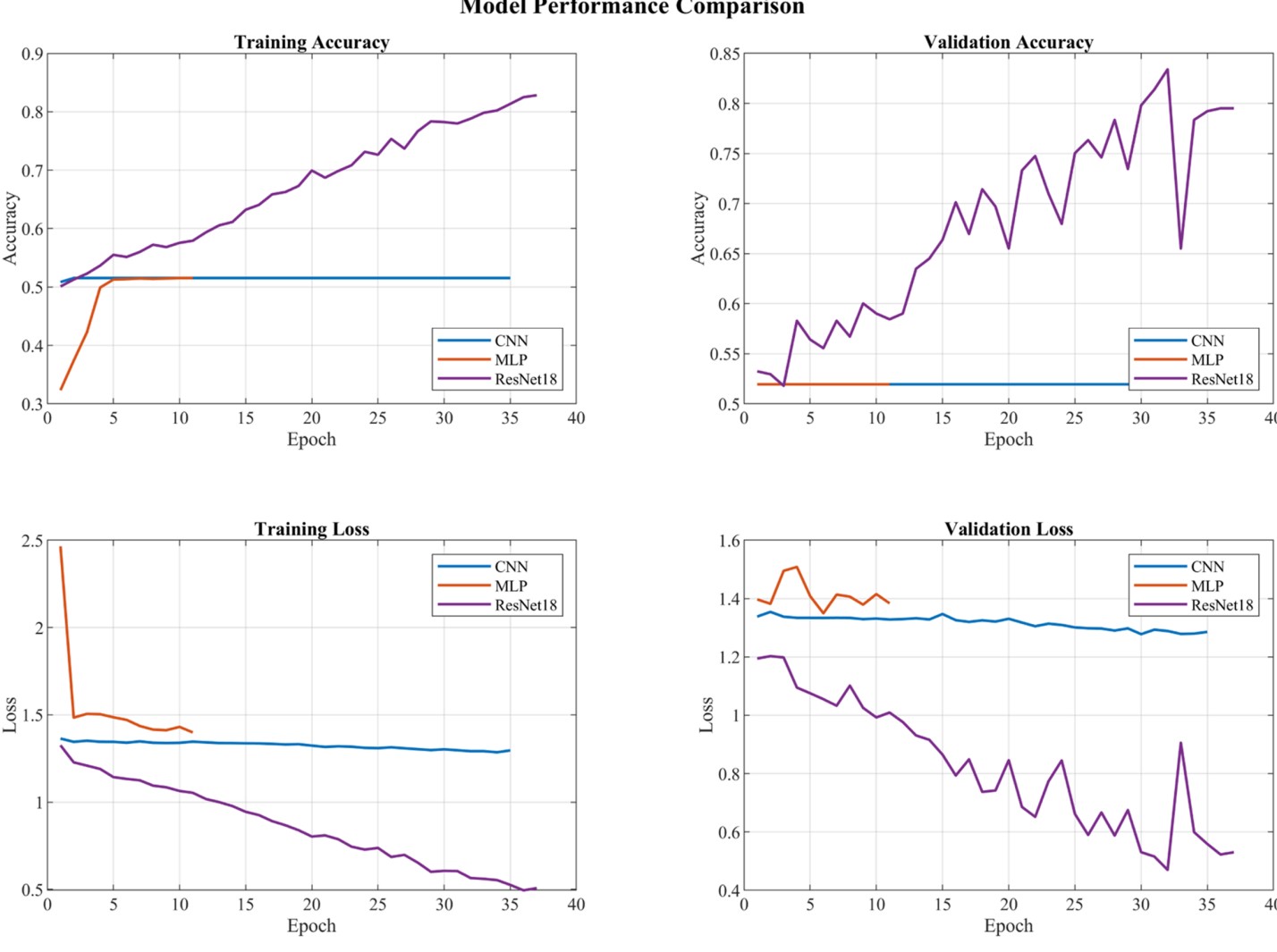

**Fig 8**. **Network architectures performance comparison.** Performance of CNN, MLP, and ResNet18. CNN/MLP val 51.95% (CNN train 51.53%); ResNet18 train 82.82%, val 83.41%.; ResNet18 performs notably better.

architectures. The conventional CNN and MLP structures showed limited classification capability, both achieving validation accuracies of only 51.95%. Specifically, the CNN's training accuracy remained at 51.53%, while the MLP showed no significant improvement throughout the training process.

ResNet18 demonstrated superior learning capability, achieving a final training accuracy of 82.82% and peak validation accuracy of 83.41%. This substantial performance gap can be attributed to ResNet18's residual connections, which facilitate better gradient flow and enable more effective feature learning. The comparative results suggest that the task's complexity requires the sophisticated architecture that ResNet18 provides, while simpler models suffer from underfitting.

## Hyperparameter optimization results

Bayesian optimization was employed to systematically explore the hyperparameter of the proposed model. The optimization process involved 30 trials with different parameter combinations, as shown in Fig 9, which illustrates the relationship

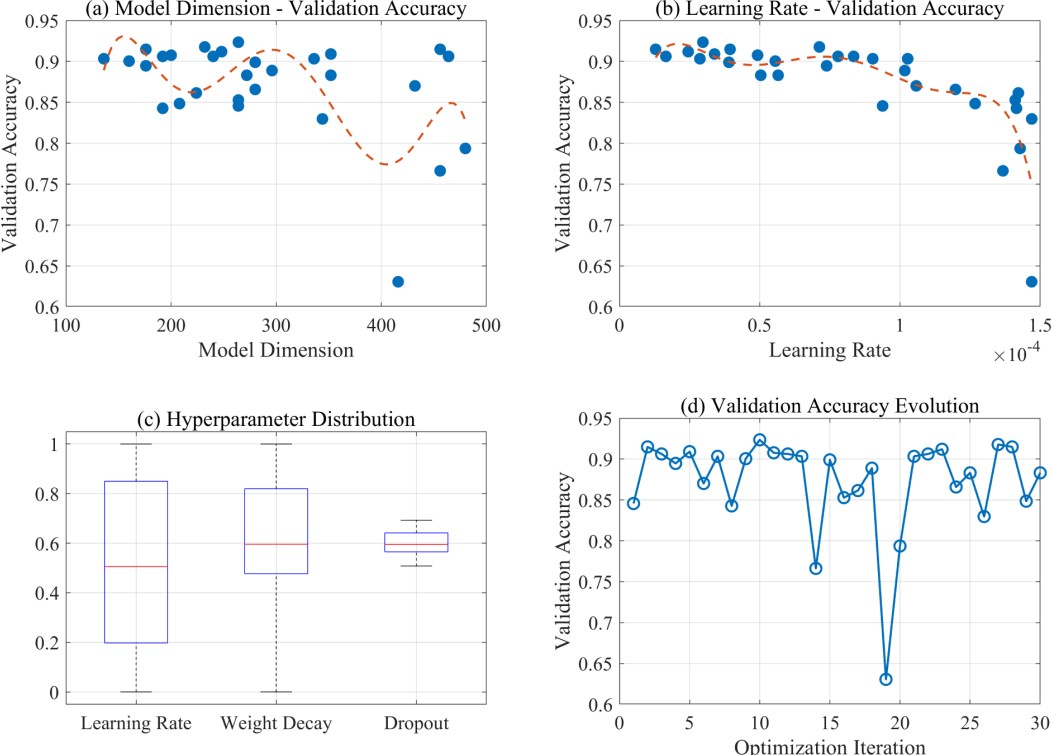

**Fig 9. Hyperparameter optimization results.** Bayesian optimization explored 30 configurations across learning rate, weight decay, dropout, model dimension (d_model), number of layers, and feedforward dimension, with nhead fixed at 8.

between key hyperparameters and validation accuracy. The hyperparameter included learning rate, weight decay, dropout rate, model dimension (d_model), number of layers, and feedforward dimension. The number of attention heads (nhead) was fixed at 8, following common practice in transformer architectures, as this configuration provides a balanced trade-off between computational efficiency and model expressiveness while maintaining sufficient parallel attention mechanisms for feature learning.

The optimal configuration achieved a validation accuracy of 92.35%, with the following hyperparameters shown in Table 1.

**Table 1**. **Optimal hyperparameters configuration.**

| Parameter | Value |
|---|---|
| d_model | 265 |
| dim_feedforward | 1335 |
| dropout | 0.61 |
| learning rate | 0.0000297 |
| num_heads | 8 |
| num_layers | 1 |
| weight_decay | 0.000987 |

The best validation accuracy (92.35%) was obtained with a shallow-but-wide transformer setting.

This configuration represents a relatively shallow but wide architecture, suggesting that for this specific task, a single transformer layer with sufficient model width is more effective than deeper architectures. The adoption of a moderate dropout rate and small learning rate enables stable training convergence while preventing overfitting.

### Final model performance

The proposed model was trained for 100 epochs with an early stopping mechanism, using the optimal hyperparameters identified through Bayesian optimization. The training process demonstrated consistent and stable improvement before automatically terminating at epoch 34. The model exhibited strong learning capability from the early stages, improving training accuracy from 53.55% to 91.99% while reducing training loss from 1.142 to 0.220 over the course of training.

The model achieved its peak validation accuracy of 94.81% at epoch 26 and maintained strong performance until early stopping was triggered, with validation loss decreasing from an initial 0.840 to 0.155 by the final epoch. Given the inherent temporal correlations in cardiac magnetometer signals, the ResNet-LSTM network architecture was selected for comparison due to its established capability in modeling sequential dependencies through LSTM units while extracting spatial features via ResNet. The comparison results are shown in Fig 10. The proposed architecture demonstrated superior performance over this reference network (92.06% validation accuracy), suggesting that transformer encoders can more effectively capture the temporal patterns in cardiac signals. This improvement, along with the consistent performance between training and validation metrics and the gradual convergence pattern, validates the architectural design while confirming the model's ability to avoid overfitting.

### Ablation studies

To validate the effectiveness of the proposed architecture, comprehensive ablation experiments were conducted comparing four model variants: ResNet without transfer learning (ResNet), ResNet with transfer learning (ResNet-TL), ResNet with transformer encoder without transfer learning (ResNet-TE), and the complete architecture (ResNet with transfer learning and transformer encoder; ResNet-TL-TE). These experiments examine both the impact of transfer learning and the contribution of transformer encoders in temporal pattern recognition.The comparative results of these ablation experiments are presented in Fig 11.

The experimental results demonstrate distinct advantages from both components. ResNet without transfer learning achieved a maximum validation accuracy of 83.41%. Adding transfer learning improved the performance to 92.35%, indicating that knowledge from natural image domains effectively enhances feature extraction capabilities. When incorporating the transformer encoder without transfer learning, the model reached 85.57% validation accuracy. This improvement over the basic ResNet demonstrates that transformer encoders' self-attention mechanism is particularly effective at capturing long-range temporal dependencies and complex sequential patterns in cardiac magnetometer signals, which is crucial given the time-series nature of CDVM data. The complete architecture, ResNet-TL-TE, achieved the highest validation accuracy of 94.81%. This superior performance suggests that the two components are complementary - transfer learning provides robust spatial feature extraction, while transformer encoders effectively model the temporal relationships between these features.

The convergence patterns reveal further insights into each component's contribution. Models with transfer learning showed faster initial convergence, likely due to the better initialized feature extractors. Meanwhile, models with transformer encoders demonstrated more stable training processes and better final performance, particularly in maintaining consistent validation metrics. This stability can be attributed to the transformer's ability to adaptively focus on relevant temporal patterns. The complete architecture achieved the lowest and most stable validation loss (0.155), suggesting that the combination of transfer learning and transformer encoder provides an effective framework for CDVM classification.

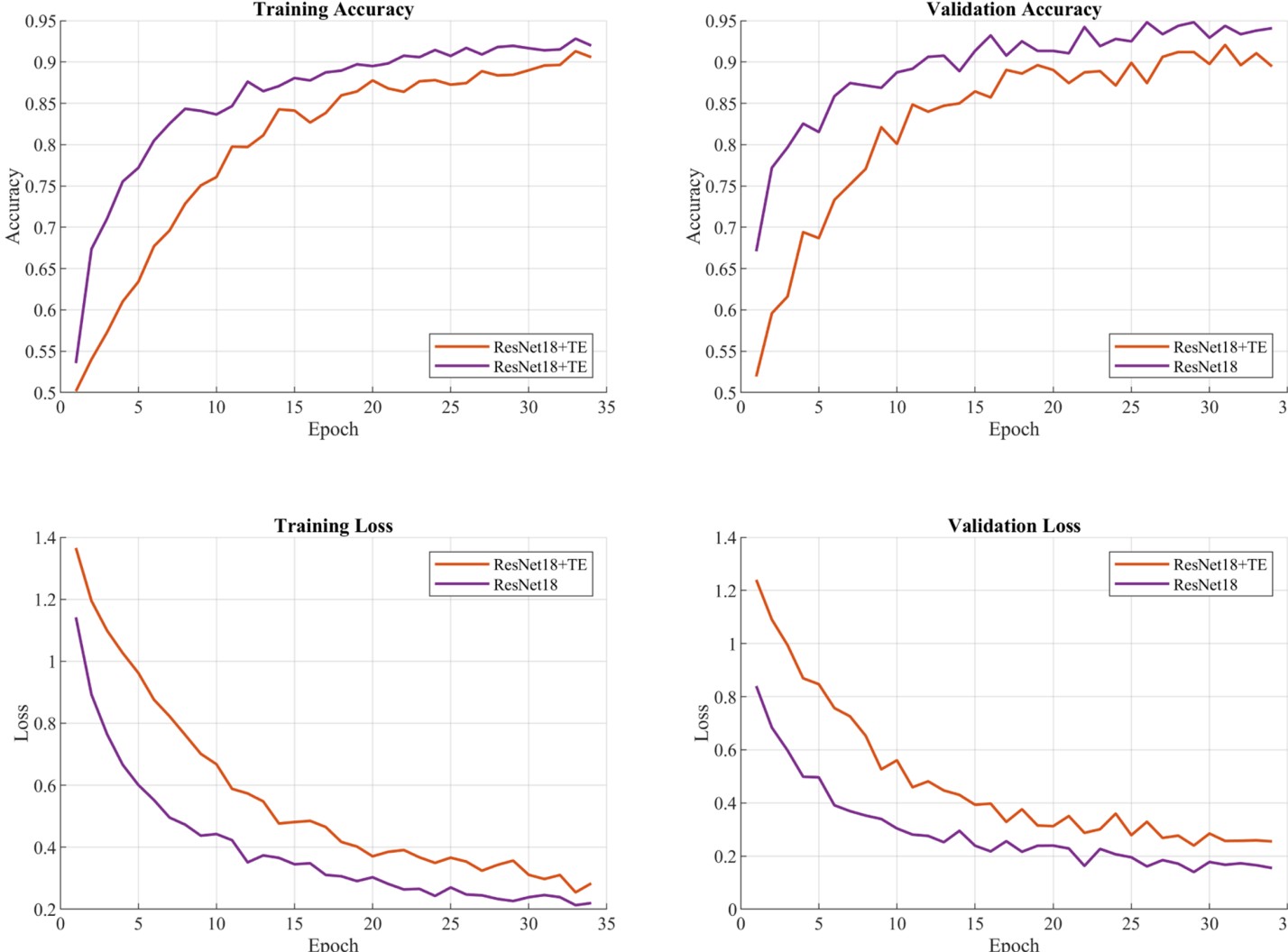

**Fig 10. ResNet18-LSTM vs ResNet18-transformer encoder.** The proposed model peaks at 94.81% validation accuracy at epoch 26, outperforming the ResNet-LSTM network (92.06%), with steadily decreasing validation loss and early stopping, indicating better temporal pattern modeling by the Transformer and no overfitting.

## Detailed performance metrics

To evaluate the model's classification capabilities comprehensively, the analysis employed multiple metrics: precision, recall, specificity, and accuracy. These metrics are defined as:

$$\text{Precision} \quad = \quad \frac{TP}{TP + FP} \tag{2}$$

$$\text{Recall} \quad = \quad \frac{TP}{TP + FN} \tag{3}$$

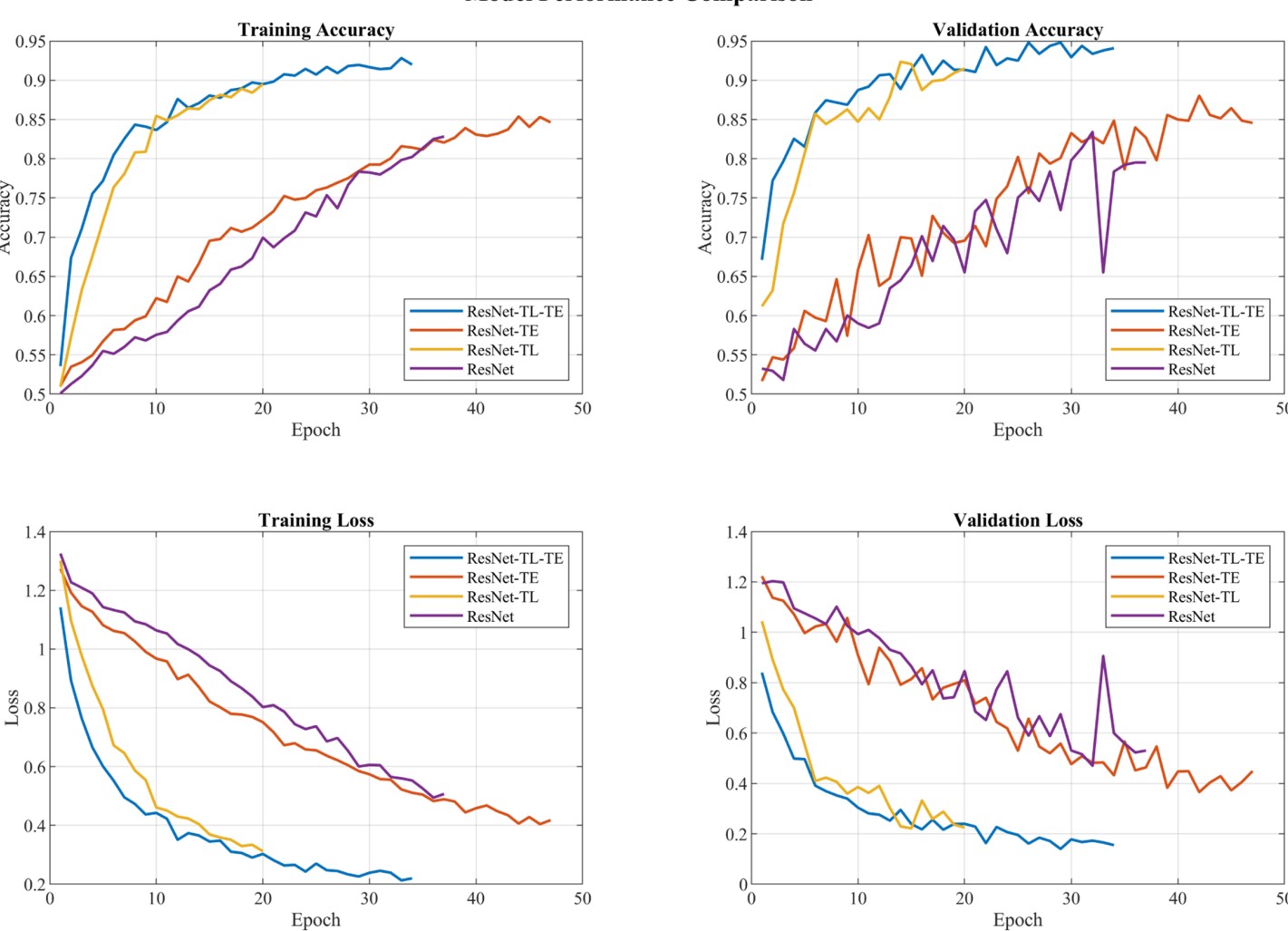

**Fig 11**. **Transfer learning and Transformer encoder components ablation studies results.** ResNet: 83.41% val. accuracy; +TL: 92.35%; +TE (no TL): 85.57%; full ResNet-TL-TE: 94.81% with the lowest, most stable validation loss (0.155). TL accelerates convergence via better initialization, while TE improves temporal modeling and training stability.

$$\text{Specificity} \quad = \quad \frac{TN}{TN + FP} \tag{4}$$

$$\text{Accuracy} \quad = \quad \frac{TP + TN}{TP + FP + TN + FN} \tag{5}$$

where TP, TN, FP, and FN denote true positives, true negatives, false positives, and false negatives, respectively. The ResNet-TL-TE model achieved an outstanding average accuracy of 97.52% on the test set. To better understand the model's performance patterns, a confusion matrix was generated from the test set results as shown in Fig 12, which

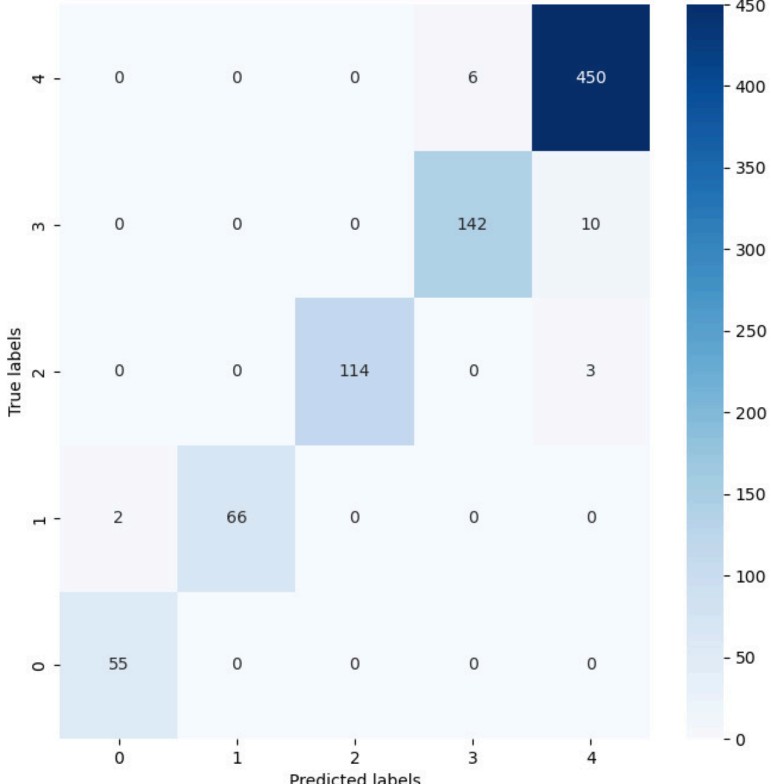

**Fig 12**. **Confusion matrix of test set.** The ResNet-TL-TE model attains an average accuracy of 97.52% on CDVM classification. The matrix visualizes per-category predictions and errors, revealing near-perfect diiesscrimination for Categories 1–2 and balanced performance across all categories.

provides a detailed visualization of the classification outcomes across different categories. The detailed performance metrics for each category on the test set are presented in Table 2.

The results demonstrate exceptional performance across all categories, with particularly strong results in Categories 1 and 2 (100% precision and specificity). Category 0 achieved perfect recall (100%), while Categories 3 and 4 maintained high performance across all metrics. These comprehensive results validate the robustness and reliability of the proposed approach for CDVM classification.

**Table 2**. **Precision, Recall, and Specificity for Each Category.**

| Category | Precision | Recall | Specificity |
|---|---|---|---|
| Category 0 | 96.49% | 100% | 99.76% |
| Category 1 | 100% | 97.06% | 100% |
| Category 2 | 100% | 97.44% | 100% |
| Category 3 | 95.95% | 93.42% | 99.28% |
| Category 4 | 97.19% | 98.68% | 98.45% |

Categories 1–2 reach 100% precision and specificity; Category 0 has 100% recall; other Categories are high overall.

## Conclusion

This paper presents a novel hybrid deep learning architecture (ResNet-TL-TE) for CDVM classification by integrating ResNet-18, transfer learning, and Transformer encoders. The experimental results demonstrate several significant contributions to the field:

The proposed ResNet-TL-TE model achieves state-of-the-art classification accuracy of 97.52%, outperforming existing approaches such as SVM and ResNet-LSTM. This approach not only effectively addresses the challenges of limited sample sizes and network degradation, but also improves classification accuracy and efficiency, providing more effective assistance to clinicians in medical diagnostic decision-making.

The comprehensive evaluation across all five categories reveals exceptional performance metrics, with precision and specificity reaching 100% in multiple categories, demonstrating the model's robust clinical applicability. Furthermore, the framework exhibits remarkable scalability, allowing for parameter and structural adaptations to accommodate increasing data volumes and more complex classification requirements.

However, there are still several issues that need to be addressed: The original dataset for this task is relatively small. Access to a larger dataset or exploring additional data augmentation techniques would allow for better optimization of network parameters, prevent overfitting, and ultimately strengthen the network's generalization capabilities. The network architecture utilized in this task demonstrates satisfactory efficiency during training, but there's still room for further optimization. Exploring alternative lightweight network architectures, like MobileNet, could potentially reduce computational complexity, parameter count, and inference time while maintaining or even enhancing accuracy.

## Supporting information

**S1 MCG raw data 1. Test 0-4.** The raw MCG dataset includes categories 0-4 for testing.
(ZIP)

**S2 MCG raw data 2. Train 0-3.** The raw MCG dataset includes categories 0-3 for training and validation.
(ZIP)

**S3 MCG raw data 3. Train 4.** The raw MCG dataset includes category 4 for training and validation.
(ZIP)

## Author contributions

**Data curation:** Yunfeng Yang, Qingxin Zeng, Xiaohu Zhao, Zhenghui Hu.

**Formal analysis:** Wenyue Yu, Xiaohu Zhao.

**Funding acquisition:** Qiang Lin, Zhenghui Hu.

**Methodology:** Lihui Zhu, Wenyue Yu.

**Supervision:** Zhan Zheng.

**Validation:** Yunfeng Yang.

**Visualization:** Lihui Zhu, Qingxin Zeng.

**Writing – original draft:** Lihui Zhu.

**Writing – review & editing:** Zhan Zheng, Qiang Lin, Zhenghui Hu.

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
