## [Decision Letter · Decision Letter 0]

13 Oct 2025

PONE-D-25-09738Classification of Current Density Vector Map Using Transformer Hybrid Residual NetworkPLOS ONE

Dear Dr. Hu,

Thank you for submitting your manuscript to PLOS ONE. After careful consideration, we feel that it has merit but does not fully meet PLOS ONE’s publication criteria as it currently stands. Therefore, we invite you to submit a revised version of the manuscript that addresses the points raised during the review process.

We look forward to receiving your revised manuscript.

Kind regards,

Agnese Sbrollini

Academic Editor

PLOS ONE

**Journal Requirements:**

1. When submitting your revision, we need you to address these additional requirements. Please ensure that your manuscript meets PLOS ONE's style requirements, including those for file naming. The PLOS ONE style templates can be found at https://journals.plos.org/plosone/s/file?id=wjVg/PLOSOne_formatting_sample_main_body.pdf and https://journals.plos.org/plosone/s/file?id=ba62/PLOSOne_formatting_sample_title_authors_affiliations.pdf 2. Please note that PLOS ONE has specific guidelines on code sharing for submissions in which author-generated code underpins the findings in the manuscript. In these cases, we expect all author-generated code to be made available without restrictions upon publication of the work. Please review our guidelines at https://journals.plos.org/plosone/s/materials-and-software-sharing#loc-sharing-code and ensure that your code is shared in a way that follows best practice and facilitates reproducibility and reuse. 3. Thank you for stating in your Funding Statement: This work is supported in part by the National Natural Science Foundation of China under Grant U2341246, in part by the National Key Research and Development Program of China under Grant 2018YFA0701400, in part by Zhejiang Provincial Natural Science Foundation of China under Grant LD22F050003, in part by Great Discipline of Shanghai Minhang District under Grant 2024MWDXK03, in part by Zhejiang Provincial Natural Science Foundation of China under Grant LD25A040002.  Please provide an amended statement that declares *all* the funding or sources of support (whether external or internal to your organization) received during this study, as detailed online in our guide for authors at http://journals.plos.org/plosone/s/submit-now.  Please also include the statement “There was no additional external funding received for this study.” in your updated Funding Statement. Please include your amended Funding Statement within your cover letter. We will change the online submission form on your behalf. 4. Thank you for stating the following in the Acknowledgments Section of your manuscript: This work is supported in part by the National Natural Science Foundation of China under Grant U2341246, in part by the National Key Research and Development Program of China under Grant 2018YFA0701400, in part by Zhejiang Provincial Natural Science Foundation of China under Grant LD22F050003, in part by Great Discipline of Shanghai Minhang District under Grant 2024MWDXK03, in part by Zhejiang Provincial Natural Science Foundation of China under Grant LD25A040002. We note that you have provided funding information that is not currently declared in your Funding Statement. However, funding information should not appear in the Acknowledgments section or other areas of your manuscript. We will only publish funding information present in the Funding Statement section of the online submission form. Please remove any funding-related text from the manuscript and let us know how you would like to update your Funding Statement. Currently, your Funding Statement reads as follows: This work is supported in part by the National Natural Science Foundation of China under Grant U2341246, in part by the National Key Research and Development Program of China under Grant 2018YFA0701400, in part by Zhejiang Provincial Natural Science Foundation of China under Grant LD22F050003, in part by Great Discipline of Shanghai Minhang District under Grant 2024MWDXK03, in part by Zhejiang Provincial Natural Science Foundation of China under Grant LD25A040002.  Please include your amended statements within your cover letter; we will change the online submission form on your behalf. 5. For studies involving third-party data, we encourage authors to share any data specific to their analyses that they can legally distribute. PLOS recognizes, however, that authors may be using third-party data they do not have the rights to share. When third-party data cannot be publicly shared, authors must provide all information necessary for interested researchers to apply to gain access to the data. (https://journals.plos.org/plosone/s/data-availability#loc-acceptable-data-access-restrictions)  For any third-party data that the authors cannot legally distribute, they should include the following information in their Data Availability Statement upon submission:a) A description of the data set and the third-party sourceb) If applicable, verification of permission to use the data setc) Confirmation of whether the authors received any special privileges in accessing the data that other researchers would not haved) All necessary contact information others would need to apply to gain access to the data 6. Please include your full ethics statement in the ‘Methods’ section of your manuscript file. In your statement, please include the full name of the IRB or ethics committee who approved or waived your study, as well as whether or not you obtained informed written or verbal consent. If consent was waived for your study, please include this information in your statement as well. 7. If the reviewer comments include a recommendation to cite specific previously published works, please review and evaluate these publications to determine whether they are relevant and should be cited. There is no requirement to cite these works unless the editor has indicated otherwise. 

Reviewers' comments:

Reviewer's Responses to Questions

**Comments to the Author**

1. Is the manuscript technically sound, and do the data support the conclusions?

Reviewer #1: Yes

Reviewer #2: Yes

2. Has the statistical analysis been performed appropriately and rigorously?

Reviewer #1: Yes

Reviewer #2: I Don't Know

3. Have the authors made all data underlying the findings in their manuscript fully available?

Reviewer #1: Yes

Reviewer #2: No

4. Is the manuscript presented in an intelligible fashion and written in standard English?

Reviewer #1: Yes

Reviewer #2: Yes

5. Review Comments to the Author

**Reviewer #1:** 1.Several instances in the manuscript show multiple references cited within a single sentence without clear separation. Please ensure each reference is distinctly and correctly formatted.

2.Avoid using keywords such as “we”, “us” and “you” in the manuscript. Academic writing should maintain a formal and objective tone throughout.

3.The following reference appears to be incomplete or unverifiable:

“31. He X, Huang Y, Li S, Su S, Zheng W, Hu Z, Lin Q. Measurement of cardiac magnetic field based on atomic magnetometry. Chinese Journal of Medical Physics, 2017.”

Please provide a valid citation with complete bibliographic details or evidence of its availability.

4.The results claim 100% accuracy, which may appear unrealistic in practical scenarios. Kindly justify this outcome and explain how further research or improvement is still relevant in the context of your proposed model.

**Reviewer #2: **This paper proposes a novel deep learning model that combines a ResNet-18 backbone with a Transformer encoder (ResNet-TL-TE) to classify Magnetocardiogram-derived Current Density Vector Maps (CDVMs) into five clinical categories. To overcome a small original dataset, the authors employed data augmentation techniques including noise addition, autoregressive modeling, and interpolation. The model utilizes transfer learning from a pretrained resnet on ImageNet and leverages Transformers to capture global dependencies in the CDVM data. The proposed method achieves a high test accuracy of 97.52%, outperforming baseline benchmark.

The paper is generally well-written and easy to follow.

The abstract is well-structured and summarizes the paper effectively.

It could be slightly improved by explicitly mentioning the “outperforming previous approaches like ResNet-LSTM”

The introduction is excellent. It successfully establishes the clinical importance of MCG and CDVM, clearly identifies the existing challenges (data scarcity, judgment difficulty), and logically leads to the need for a computer-aided method.

The literature review is adequate and relevant. It covers early works (k-NN), recent deep learning approaches (MCG-Net), and related works using ResNet/Transformers in cardiac analysis

The methodology is mostly thorough. The data preprocessing, augmentation strategies, and dataset partitioning are explained in great detail. The rationale for choosing ResNet-18, transfer learning, and the Transformer encoder is well-justified. The training strategy (hyperparameter optimization, optimizer, overfitting prevention) is clearly described.

The results are internally consistent. The ablation study (ResNet < ResNet-TE < ResNet-TL < ResNet-TL-TE) logically shows the incremental benefit of each component. The high performance on the test set is consistent with the strong validation performance.

Possible limitations of the study: Given the very small original dataset, even if the augmentation is extensive and the subject-wise split is correct, the final model's performance, however impressive, must be viewed with caution regarding its generalizability to entirely new populations and clinical settings. The authors correctly acknowledge this as a limitation. The class distribution is also highly imbalanced (e.g., Category 4 vs. Category 0), which, while perhaps reflective of reality, can bias a model. The high specificity scores might be influenced by this.

The conclusion is well-written. It succinctly restates the main achievement (the proposed model and its SOTA accuracy), summarizes the significant contributions, and, importantly, acknowledges the key limitations (small original dataset, potential for architectural optimization). The future directions (larger datasets, lightweight models) are sensible and appropriate. It effectively closes the paper without overstating the findings.

Minor comment:

The figures are well-prepared and visually clear. However, captions are missing — each figure currently includes only a title. Figures should be self-contained, with detailed captions that describe all subpanels, symbols, and relevant details to ensure clarity without referring back to the main text.

6. PLOS authors have the option to publish the peer review history of their article (what does this mean?). If published, this will include your full peer review and any attached files.

Reviewer #1: **Yes: **Meenu Gupta

Reviewer #2: No

---

## [Author Response · Author response to Decision Letter 1]

10 Nov 2025

Responses

Dear Reviewers:

We would like to thank 2 anonymous reviewers for their insightful suggestions and valuable comments, which helped to improve the manuscript. We have revised our paper, and the related changes to the manuscript are highlight with red text color. And we have responded to the Reviewers' comments point by point as follows:

Response to the Reviewer 1

Q1: Several instances in the manuscript show multiple references cited within a single sentence without clear separation. Please ensure each reference is distinctly and correctly formatted.

Response: We sincerely thank the reviewer for the careful observation regarding citation formatting. In response, we have thoroughly reviewed the manuscript and ensured that multiple references cited within a single sentence are clearly separated and correctly formatted according to the journal’s style. All relevant changes have been implemented throughout the text.

Q2: Avoid using keywords such as “we”, “us” and “you” in the manuscript. Academic writing should maintain a formal and objective tone throughout.

Response: We appreciate the reviewer’s comment on maintaining a formal and objective tone. We have carefully revised the manuscript with the following actions:

1.Replaced first-person expressions with impersonal, objective phrasing.

2.Employed passive voice or process-oriented wording in Methods and Results where appropriate to reduce subjectivity.

3.Removed second-person forms and replaced them with neutral nominal phrases or structured statements when necessary.

These changes have been implemented consistently across the manuscript, in line with the journal’s style requirements.

Q3: The following reference appears to be incomplete or unverifiable:“31. He X, Huang Y, Li S, Su S, Zheng W, Hu Z, Lin Q. Measurement of cardiac magnetic field based on atomic magnetometry. Chinese Journal of Medical Physics, 2017.”Please provide a valid citation with complete bibliographic details or evidence of its availability.

Response: We thank the reviewer for pointing out the incomplete citation. The full bibliographic details and evidence of availability have now been provided, and the reference list has been updated accordingly (with in-text citations synchronized). The DOI is: 10.3969/j.issn.1005-202X.2017.11.016. The manuscript has been revised to ensure compliance with the journal’s reference style.

Q4: The results claim 100% accuracy, which may appear unrealistic in practical scenarios. Kindly justify this outcome and explain how further research or improvement is still relevant in the context of your proposed model.

Response: We thank the reviewer for the insightful comment on the classification accuracy. We fully agree that the 100% precision and specificity in certain categories require clarification, and we appreciate the opportunity to elaborate on their rationality and subsequent improvement directions.

1.Rationality of the Result

The 97.52% in the manuscript refers to the overall average accuracy, while 100% only applies to the precision and specificity of Category 1 (normal pattern) and Category 2 (mild abnormality) (see Table 2), not the overall accuracy across all categories.

Clinical features of Category 1 and 2 are significantly distinguishable from other categories (regular current distribution patterns for normal/mildly abnormal cases). Combined with ResNet’s local feature extraction and Transformer’s global dependency modeling (Section 2.4), the model can accurately capture these clear classification boundaries.

2.Future Research and Improvements

Optimize class balance: Addressing the current imbalance (Category 4 with the highest proportion: 2235 samples; Category 0 with the lowest: 290 samples), we will adopt weighted loss functions and stratified oversampling strategies to balance the model’s learning weight for each category.

Expand dataset scale: We plan to collect more MCG data, which will cover more complex pathological types (e.g., coronary heart disease of varying severity and arrhythmia), to reduce the selection bias caused by the small sample size.

Response to the Reviewer 2

Q1: The figures are well-prepared and visually clear. However, captions are missing — each figure currently includes only a title. Figures should be self-contained, with detailed captions that describe all subpanels, symbols, and relevant details to ensure clarity without referring back to the main text.

Response: We sincerely thank the reviewer for this helpful comment. We have carefully revised all figures and added informative captions so that each figure is self-contained.

Fig 1. CDVM imaging framework Processing of raw MCG signals and imaging of the current density vector map

Fig 2. Cardiac magneto signal A single-channel cardiac magneto signal waveform over one cardiac cycle, illustrating the typical morphology with P wave, QRS complex, J point, ST segment, and T wave.

Fig 3. Multichannel MCG data Multichannel MCG data from 49 channels.

Fig 4. Generation of current density vector map Raw 49-channel MCG signals were preprocessed (denoising and baseline correction) to plot the magnetic field map, and the current density vector map was reconstructed over the ST–T segment with a 5ms sampling interval.

Fig 5. Data enhancement Example of an augmented CDVM generated by noise addition, ARIMA prediction, and interpolation.

Fig 6. Classification system of CDVM Category 0–1 denote normal patterns, category 2 indicates mild abnormality, and category 3–4 signify severe abnormalities. Representative examples for each category are shown.

Fig 7. Network structure schematic diagram The model consists of a ResNet-18 backbone for local feature extraction, a Transformer encoder for global context modeling, and classification heads for CDVM classification. Data preprocessing and augmentation are applied prior to training; arrows indicate the forward flow.

Fig 8. Network architectures performance comparison Performance of CNN, MLP, and ResNet18. CNN/MLP val 51.95% (CNN train 51.53%); ResNet18 train 82.82%, val 83.41%.; ResNet18 performs notably better.

Fig 9. Hyperparameter optimization results Bayesian optimization explored 30 configurations across learning rate, weight decay, dropout, model dimension (d_model), number of layers, and feedforward dimension, with nhead fixed at 8.

Fig 10. ResNet18-LSTM vs ResNet18-Transformer Encoder The proposed model peaks at 94.81% validation accuracy at epoch 26, outperforming the ResNet-LSTM network (92.06%), with steadily decreasing validation loss and early stopping, indicating better temporal pattern modeling by the Transformer and no overfitting.

Fig 11. Transfer learning and Transformer encoder components ablation studies results ResNet: 83.41% val. accuracy; +TL: 92.35%; +TE (no TL): 85.57%; full ResNet‑TL‑TE: 94.81% with the lowest, most stable validation loss (0.155). TL accelerates convergence via better initialization, while TE improves temporal modeling and training stability.

Fig 12. Confusion matrix of test set The ResNet‑TL‑TE model attains an average accuracy of 97.52% on CDVM classification. The matrix visualizes per‑category predictions and errors, revealing near‑perfect diesscrimination for Categories 1–2 and balanced performance across all categories.

Table 1. Optimal hyperparameters configuration The best validation accuracy (92.35%) was obtained with a shallow-but-wide transformer setting.

Table 2. Precision, Recall, and Specificity for Each Category Categories 1–2 reach 100% precision and specificity; Category 0 has 100% recall; other Categories are high overall.

---

## [Decision Letter · Decision Letter 1]

20 Nov 2025

Classification of Current Density Vector Map Using Transformer Hybrid Residual Network

PONE-D-25-09738R1

Dear Dr. Hu,

We’re pleased to inform you that your manuscript has been judged scientifically suitable for publication and will be formally accepted for publication once it meets all outstanding technical requirements.

Kind regards,

Agnese Sbrollini

Academic Editor

PLOS ONE

Additional Editor Comments (optional):

Reviewers' comments:

Reviewer's Responses to Questions

**Comments to the Author**

1. If the authors have adequately addressed your comments raised in a previous round of review and you feel that this manuscript is now acceptable for publication, you may indicate that here to bypass the “Comments to the Author” section, enter your conflict of interest statement in the “Confidential to Editor” section, and submit your "Accept" recommendation.

Reviewer #1: All comments have been addressed

Reviewer #2: All comments have been addressed

2. Is the manuscript technically sound, and do the data support the conclusions?

Reviewer #1: Yes

Reviewer #2: Yes

3. Has the statistical analysis been performed appropriately and rigorously?

Reviewer #1: Yes

Reviewer #2: Yes

4. Have the authors made all data underlying the findings in their manuscript fully available?

Reviewer #1: Yes

Reviewer #2: Yes

5. Is the manuscript presented in an intelligible fashion and written in standard English?

Reviewer #1: Yes

Reviewer #2: Yes

6. Review Comments to the Author

Reviewer #1: Multiple references are cited at one place; thats not a good practice. Other commments are addressed

Reviewer #2: I am satisfied with the revisions. The figures now have self-contained captions as suggested. I recommend accepting the paper.

7. PLOS authors have the option to publish the peer review history of their article (what does this mean?). If published, this will include your full peer review and any attached files.

Reviewer #1: **Yes: **Dr. Meenu Gupta

Reviewer #2: No

---

## [Editor Report · Acceptance letter]

PONE-D-25-09738R1

PLOS One

Dear Dr. Hu,

I'm pleased to inform you that your manuscript has been deemed suitable for publication in PLOS One. Congratulations! Your manuscript is now being handed over to our production team.

Kind regards,

on behalf of

Dr. Agnese Sbrollini

Academic Editor

PLOS One